# Comparison of Protective Effects of Antidepressants Mediated by Serotonin Receptor in Aβ-Oligomer-Induced Neurotoxicity

**DOI:** 10.3390/biomedicines12061158

**Published:** 2024-05-23

**Authors:** Ken Yamamoto, Mayumi Tsuji, Tatsunori Oguchi, Yutaro Momma, Hideaki Ohashi, Naohito Ito, Tetsuhito Nohara, Tatsuya Nakanishi, Atsushi Ishida, Masahiro Hosonuma, Toru Nishikawa, Hidetomo Murakami, Yuji Kiuchi

**Affiliations:** 1Department of Pharmacology, Showa University Graduate School of Medicine, Tokyo 142-8555, Japan; ken_yamamoto@med.showa-u.ac.jp (K.Y.); t.oguchi@med.showa-u.ac.jp (T.O.); t.nkns@med.showa-u.ac.jp (T.N.); a.ishida3929@gmail.com (A.I.); masa-hero@med.showa-u.ac.jp (M.H.); tnis.psyc@tmd.ac.jp (T.N.); ykiuchi@med.showa-u.ac.jp (Y.K.); 2Department of Neurology, Showa University School of Medicine, Tokyo 142-8666, Japan; m-yutaro@med.showa-u.ac.jp (Y.M.); mimixart@yahoo.co.jp (H.O.); hitto226@med.showa-u.ac.jp (N.I.); t-nohara2@med.showa-u.ac.jp (T.N.); hidneu@med.showa-u.ac.jp (H.M.); 3Pharmacological Research Center, Showa University, Tokyo 142-8555, Japan

**Keywords:** Alzheimer’s disease, amyloid β oligomer, antidepressants, antioxidant, 5-HT_1A_ receptor

## Abstract

Amyloid β-peptide (Aβ) synthesis and deposition are the primary factors underlying the pathophysiology of Alzheimer’s disease (AD). Aβ oligomer (Aβo) exerts its neurotoxic effects by inducing oxidative stress and lesions by adhering to cellular membranes. Though several antidepressants have been investigated as neuroprotective agents in AD, a detailed comparison of their neuroprotection against Aβo-induced neurotoxicity is lacking. Here, we aimed to elucidate the neuroprotective effects of clinically prescribed selective serotonin reuptake inhibitors, serotonin–norepinephrine reuptake inhibitors, and noradrenergic and specific serotonergic antidepressants at the cellular level and establish the underlying mechanisms for their potential clinical applications. Therefore, we compared the neuroprotective effects of three antidepressants, fluoxetine (Flx), duloxetine (Dlx), and mirtazapine (Mir), by their ability to prevent oxidative stress-induced cell damage, using SH-SY5Y cells, by evaluating cell viability, generation of reactive oxygen species (ROS) and mitochondrial ROS, and peroxidation of cell membrane phospholipids. These antidepressants exhibited potent antioxidant activity (Dlx > Mir > Flx) and improved cell viability. Furthermore, pretreatment with a 5-hydroxytryptamine 1A (5-HT_1A_) antagonist suppressed their effects, suggesting that the 5-HT_1A_ receptor is involved in the antioxidant mechanism of the antidepressants’ neuroprotection. These findings suggest the beneficial effects of antidepressant treatment in AD through the prevention of Aβ-induced oxidative stress.

## 1. Introduction

Alzheimer’s disease (AD), a progressive neurodegenerative condition and the most frequent cause of dementia, is characterized by cognitive impairment among older adults. The World Health Organization (WHO) estimates indicate over 55 million individuals living with dementia currently, and approximately 10 million new cases are diagnosed each year. AD accounts for approximately 60–70% of all dementia cases [1]. Pathologically, AD is characterized by the presence of senile plaques formed by amyloid β-peptide (Aβ) and the neurofibrillary tangles composed of hyperphosphorylated tau protein within the brain’s hippocampus and cerebral cortex regions. Accumulating these protein aggregates has been shown to cause damage to neurons and synapses, resulting in memory impairment and various cognitive dysfunctions [2]. Although the precise etiology of AD remains elusive, a range of hypotheses exists, among which is one positing Aβ as a potential causative factor. The widely accepted amyloid cascade hypothesis posits that the build-up of Aβ in the brain begins several decades before the onset of AD symptoms, and this accumulation initiates a cascade of events leading to synaptic dysfunction, tau-related neuronal injury, brain atrophy, and cognitive decline [3]. Aβ is generated through the cleavage of the cell-membrane-bound amyloid precursor protein in neurons, initially forming monomers. As Aβ accumulates, it aggregates in β-sheet-rich structures, which undergo structural changes and transition into dimers, trimers, tetramers, oligomers, and eventually fibrils, contributing to the formation of senile plaques. While fibrils were previously believed to be the complete aggregates possessing the greatest neurotoxicity, recent research has indicated that high-molecular-weight Aβ oligomer (Aβo), the intermediate stages of aggregation, are the most toxic form. These Aβo induce neurotoxic effects through various mechanisms, including damage to the cell membrane, interaction with membrane receptors, oxidative stress, the influx of calcium ions (Ca^2+^) into the cell, and impairment of mitochondrial function [4]. Aβo are now recognized as major contributors to the development of AD pathology, leading to synaptic loss and neuronal death within the AD brain [5,6].

While cognitive impairments such as memory loss are the characteristic clinical features, patients with AD frequently develop neuropsychiatric symptoms, also known as behavioral and psychological symptoms of dementia (BPSD). BPSD in patients with AD includes aberrant motor behavior, agitation/aggression, irritability, anxiety, depression/dysphonia, delusion, apathy, appetite abnormalities, sleep changes, disinhibition, hallucinations, and elation/euphoria [7]. Complications arising from AD can accelerate cognitive decline, significantly reducing the quality of life for patients. Therefore, it is crucial to manage these conditions diligently. Clinical studies reveal a prevalence of ‘major depressive episodes’ in patients with AD of 20–25%, with other depressive syndromes, including minor depression, affecting an additional 20–30% of patients [8,9]. A meta-analysis involving 13 studies revealed that individuals with a history of depression have approximately double the relative risk of developing AD [10].

The cognitive benefits of antidepressants in patients with dementia, including AD, have yielded inconsistent evidence. The use of antidepressants in patients with AD is not yet strongly recommended unless distinct depressive symptoms are diagnosed. Existing reviews have produced inconclusive results due to a lack of data regarding the potential benefits of antidepressant use in patients with dementia [11]. Furthermore, some studies suggest that conventional antidepressants are ineffective in depression accompanied by AD [12], emphasizing the urgent need to understand the mechanisms of antidepressant resistance. Therefore, the focus of this study was the effects of antidepressant treatment on AD. It is imperative to investigate how antidepressants compare in terms of their ability to disrupt the pathogenic mechanisms of AD and slow down the neurodegenerative processes brought on by Aβo. In addition, according to the NIH website, the most commonly prescribed antidepressants currently are SSRIs [13]. Furthermore, research comparing SSRIs and tricyclic antidepressants suggests that SSRIs have higher tolerability [14]. Therefore, we have decided to opt for newer-mechanism antidepressants currently widely used in clinical practice. In this study, we selected three antidepressant drugs: fluoxetine, Flx (an SSRI); duloxetine, Dlx (a serotonin–norepinephrine reuptake inhibitor, SNRI); and mirtazapine, Mir (a noradrenergic and specific serotonergic antidepressant, NaSSA), to compare the mechanisms of antidepressant treatments.

In a study using human neuroblastoma cells, Flx was found to reduce intracellular Aβo concentration [15]. In contrast, mouse model studies on AD have shown that Flx can lower Aβ levels in brain tissue, cerebrospinal fluid, and serum, delaying cognitive decline [16,17]. Additionally, Mir has been reported to reverse neurite atrophy caused by Aβo [18], and Flx and Dlx have demonstrated antioxidant effects in animal brain tissue [19,20]. Thus, both in vivo and in vitro studies have indicated that antidepressants may be effective against Aβo-induced toxicity and provide protective effects on brain tissue. However, it should be noted that while there is a strong body of research on the efficacy of Flx, a representative SSRI, data on other newer antidepressants and their comparison under the same conditions remain insufficient. Furthermore, the classes of antidepressants (SSRI, SNRI, and NaSSA) investigated in this study alter the serotonergic activity in the central nervous system. These antidepressants act by inhibiting the reuptake of 5-hydroxytryptamine (5-HT), also known as serotonin, thereby increasing 5-HT activity [21]. Among the 5-HT receptors targeted by 5-HT, the 5-HT_1A_ receptor is highly expressed in the brain, particularly in the hippocampus and olfactory epithelium. However, some postmortem studies on AD have indicated a decrease in 5-HT_1A_ receptor binding [22].

This study elucidates the neuroprotective effects of clinically prescribed SSRIs, SNRIs, and NaSSAs at the cellular level and establishes the underlying mechanisms for potential clinical applications. The study aimed to compare the neuroprotective effects of Flx, Dlx, and Mir against Aβ-oligomer-induced toxicity through their antioxidant activity. Additionally, our goal was to determine whether the 5-HT_1A_ receptor mediates the cellular protective effects.

## 2. Materials and Methods

### 2.1. Drugs and Reagents

Aβ_1-42_ peptides were obtained from Peptide Institute Inc. (Osaka, Japan). Flx, Dlx, Mir (Figure 1), and yohimbine hydrochloride, a selective α_2_-adrenergic receptor antagonist, were purchased from FUJIFILM Wako Pure Chemical Corporation (Osaka, Japan). WAY-100635, a selective 5-HT_1A_ receptor antagonist, was purchased from Sigma-Aldrich (St. Louis, MO, USA). All other chemicals used were of the highest commercially available purity.

### 2.2. Preparation of Aβo

We isolated and collected Aβo using a previously reported method [23,24]. In brief, Aβ_1-42_ peptides were dissolved in 10 mM NaOH and sonicated for 3 min. Phosphate-buffered saline (PBS) was added to the solution and filtered through a Millex^®^-LG filter (0.20 µm; Millipore Ireland BV, Dublin, Ireland). The filtered solution was incubated at 37 °C for 1 h, followed by centrifugation at 16,000× *g* for 5 min. The supernatant, containing Aβo, was collected, and the protein concentration was determined using a Bio-Rad Protein Assay Dye Reagent Concentrate (Bio-Rad Laboratories, Inc., Hercules, CA, USA). The Aβo solution was diluted with PBS to a concentration of 50 μM and stored at −80 °C. The Aβo solution was confirmed for the presence of Aβo using size exclusion chromatography (SEC) and transmission electron microscopy (TEM). A Superdex 75 increase 10/300 GL column (Cytiva, Tokyo, Japan) was employed, and the supernatant was fractioned at a flow rate of 0.8 mL/min using 10 mM phosphate buffer at pH 7.4. The Aβo peak was observed at 10 min as shown in Appendix A. Furthermore, we validated the morphology of Aβo using an H-7600 transmission electron microscope (TEM; Hitachi, Ltd., Tokyo, Japan), observing high-molecular-weight Aβo containing numerous protofibrils in the Aβo solution (Appendix A).

### 2.3. Cell Culture

D-MEM/Ham’s F-12 medium and all-trans retinoic acid (ATRA) were purchased from FUJIFILM Wako Pure Chemical Corporation. Streptomycin sulfate, amphotericin B, fetal bovine serum (FBS), and penicillin G sodium were all obtained from Thermo Fisher Scientific K.K. (Waltham, MA, USA).

SH-SY5Y cells (human neuroblastoma, EC-94030304) were sourced from the European Collection of Authenticated Cell Cultures (London, UK). These cells were cultured in D-MEM/Ham’s F-12 medium supplemented with 10% FBS, streptomycin sulfate, penicillin G sodium, and amphotericin B at 37 °C under a 5% CO_2_/95% air atmosphere. The cells were treated with 10 μM ATRA for 7 days to induce cellular differentiation. Flx and WAY-100635 were dissolved in the culture medium. Dlx and Mir were dissolved in dimethyl sulfoxide (DMSO) and then diluted in the medium to a final concentration of 0.1%.

### 2.4. Cell Viability Assay

#### MTT Assay

Cell viability was assessed using the 3-(4, 5-dimethylthiazoyl-2-yl)-2-5, diphenyltetrazolium bromide (MTT) assay using the MTT Cell Count Kit (Nacalai Tesque, Inc., Kyoto, Japan). The MTT assay converts colorless MTT to blue formazan by active mitochondrial dehydrogenases present only in viable cells. SH-SY5Y cells were seeded at a 5.0 × 10^5^ cells/mL density into 96-well collagen-coated plates and incubated at 37 °C for 24 h. Subsequently, the cells were exposed to 5 μM Aβo and simultaneously treated with 1.0 μM antidepressants (Flx, Dlx, and Mir). After 24 h of treatment, the formazan product was detected at 540 nm using a microplate reader (Spectra Max i3; Molecular Devices, Sunnyvale, CA, USA). In addition, SH-SY5Y cells were pretreated with 10 μM WAY-100635 or 1 μM yohimbine for 1 h and incubated with antidepressants and Aβo for 24 h before viability assays were performed.

### 2.5. Oxidative Stress Assay

#### 2.5.1. Detection of Reactive Oxygen Species (ROS)

The production of ROS resulting from Aβo exposure was examined using CM-H2DCFDA (General Oxidative Stress Indicator; Thermo Fisher Scientific K.K., Waltham, MA, USA). SH-SY5Y cells were incubated at 37 °C for 24 h at a density of 1.0 × 10^6^ cells/mL, exposed to 5 μM Aβo, and simultaneously treated with the antidepressants (1.0 μM Flx, Dlx, and Mir) for 24 h. The fluorescence emitted from ROS was measured at an excitation wavelength of 488 nm and an emission wavelength of 525 nm using a microplate reader (Spectra Max i3; Molecular Devices). The protein concentration in the samples was determined using Bio-Rad Protein Assay Dye Reagent Concentrate (Bio-Rad Laboratories, Inc.). In addition, SH-SY5Y cells were pretreated with 10 μM WAY-100635 for 1 h and incubated with the antidepressants and Aβo for 24 h before ROS measurements were performed. After fluorescence measurements, the morphology of individual cells was examined by imaging them with a fluorescence microscope (BZX800; Keyence, Osaka, Japan).

#### 2.5.2. Detection of Mitochondrial ROS

Mitochondria are known to be the main producers of ROS in SH-SY5Y cells. The mitochondrial ROS (mito-ROS) generation resulting from exposure to Aβo was measured using the Mitochondrial ROS Detection Kit (Cayman Chemical Company, Ann Arbor, MI, USA). SH-SY5Y cells were cultured at a density of 1.0 × 10^6^ cells/mL, incubated at 37 °C for 24 h, and exposed to 5 μM Aβo and the three antidepressants for 24 h. Mito-ROS levels were also measured after pretreatment of the cells with 10 µM WAY-100635. The fluorescence intensity, indicative of mito-ROS levels, was measured using a microplate reader (Spectra Max i3; Molecular Devices) at an excitation wavelength of 500 nm and an emission wavelength of 580 nm. After fluorescence measurements, the cellular protein content in the plate was determined using Bio-Rad Protein Assay Dye Reagent Concentrate (Bio-Rad Laboratories, Inc.), and the results were expressed as fluorescence intensity per unit protein.

#### 2.5.3. Phospholipid Peroxidation in Cell Membranes

Diphenyl-1-pyrenylphosphine (DPPP) is a compound that reacts with hydroperoxides and produces a fluorescent phosphine oxide. However, it remains nonfluorescent until it undergoes oxidation. SH-SY5Y cells at a 1.0 × 10^6^ cells/mL density were incubated at 37 °C for 24 h. The cells were treated with 5 μM DPPP (Thermo Fisher Scientific K.K., Waltham, MA, USA) for 10 min at 37 °C to examine the peroxidation of phospholipids in cell membranes. After the DPPP treatment, the cells were exposed to 5 μM Aβo and simultaneously treated with the three antidepressants for 30 min. In addition, we conducted measurements with 10 μM WAY-100635 pretreatment for 10 min. The fluorescence intensity of the oxidized form of DPPP was measured at the excitation wavelength of 351 nm and emission wavelength of 380 nm using a microplate reader (Spectra Max i3; Molecular Devices). The protein concentration in the samples was determined using Bio-Rad Protein Assay Dye Reagent Concentrate (Bio-Rad Laboratories, Inc.).

#### 2.5.4. Human Heme Oxygenase 1 (HO-1) Levels

HO-1 expression is induced by heat, oxidative stress, and inflammatory cytokines. The increased expression of HO-1 is considered one of the primary mechanisms for safeguarding cells against oxidative stress. For measuring the HO-1 levels, SH-SY5Y cells were cultured at a density of 1.0 × 10^6^ cells/mL, incubated at 37 °C for 24 h, and exposed to 5 μM Aβo and simultaneously treated with the antidepressants (1.0 μM Flx, Dlx, and Mir) for 24 h. HO-1 levels were also measured in cells pretreated with 10 µM WAY-100635 for 1h. The amount of HO-1 present in cell lysates was determined using an enzyme-linked immunosorbent assay (ELISA) method with a monoclonal antibody (Human Heme Oxygenase 1 (HO-1) SimpleStep ELISA Kit; ab 207621, Abcam, Cambridge, UK). The absorbance was measured at 450 nm using a microplate reader (Spectra Max i3; Molecular Devices). The protein concentration in the cell lysate was also determined using the Bio-Rad Protein Assay Dye Reagent Concentrate (Bio-Rad Laboratories, Inc.).

#### 2.5.5. Manganese Superoxide Dismutase (Mn-SOD) Levels

The main antioxidant enzymes in cells are the members of the superoxide dismutase (SOD) family, which convert superoxide into hydrogen peroxide, subsequently eliminated by other cellular defenses like glutathione peroxidase and catalase. Among the three types of SODs, SOD2 is specifically found in the mitochondria and utilizes manganese as a cofactor. In this study, SH-SY5Y cells were cultured at a density of 1.0 × 10^6^ cells/mL and incubated at 37 °C for 24 h, exposed to 5 μM Aβo, and simultaneously treated with the antidepressants (0.1 μM Flx, Dlx, and Mir) for 24 h. Mn-SOD levels were also measured in cells pretreated with 10 µM WAY-100635.

The amount of SOD2 isozyme in cell lysates was determined using an ELISA method with a monoclonal antibody (Human SOD2 ELISA Kit; ab178012, Abcam). The absorbance was measured at 450 nm using a microplate reader (Spectra Max i3; Molecular Devices). The protein concentration in the cell lysate was also determined using the Bio-Rad Protein Assay Dye Reagent Concentrate (Bio-Rad Laboratories, Inc.).

### 2.6. Statistical Analyses

Each measurement was conducted at least three times to ensure accuracy and reproducibility. The results were presented as mean ± standard error of the mean (SEM). Statistical comparisons were made between SH-SY5Y cells exposed to 5 μM Aβo with or without 0.1% DMSO to assess the effects of each antidepressant. Analysis of variance (ANOVA) was performed, followed by Tukey’s or Dunnett’s post hoc test. A *p*-value of less than 0.05 was considered a significant difference for all tests.

## 3. Results

### 3.1. Effects of Antidepressants on Aβo-Induced Cytotoxicity

#### Detection of Cell Viability by MTT Assay

Initially, the MTT assay was performed in a preliminary experiment by exposing SH-SY5Y cells to 1, 5, and 10 μM Aβo, with or without 0.1% DMSO, for 24 h. We found that compared with that of the control cells (no exposure), the cell viability reduced significantly (*p* < 0.0001) for all three concentrations of Aβo (Figure 2A,B). Based on the outcome of this preliminary experiment, an exposure concentration of 5 µM Aβo was used for all subsequent experiments. Treatment with 1.0 μM Flx showed a significant increase (*p* = 0.0061) in viability compared to 5 μM Aβo, whereas for both Dlx and Mir, treatment with 0.1 μM (in the presence of 0.1% DMSO) significantly suppressed the Aβo-induced decrease in cell viability (Dlx: *p* = 0.0083, Mir: *p* = 0.0151). Thus, the reduction in cell viability induced by Aβo was significantly reduced at lower concentrations of Dlx and Mir compared to Flx (Figure 2C,D). As all three antidepressants significantly increased cell viability at 1.0 μM, we used this concentration for the subsequent experiments.

Next, to ascertain whether the protective effects of the antidepressants against Aβo-induced cytotoxicity are mediated by 5-HT or α_2_-adrenergic receptors, the effects of 5-HT and norepinephrine (NE), as well as their respective inhibitors, WAY-100635 and yohimbine (with or without 1 h pretreatment), on cell viability, were compared. First, we decided to investigate the direct effects of administering 5-HT and NE by directly treating them, as synaptic levels of 5-HT increase with Flx treatment and both 5-HT and NE increase with Dlx and Mir treatments.

Treatment with 5-HT + Aβo showed a significant increase in viability, as opposed to the reduction in cell viability induced by Aβo alone. However, NE treatment of SH-SY5Y cells did not significantly affect the Aβo-induced reduction in viability. Moreover, the WAY-100635 pretreatment significantly impacted the cell viability increase caused by 5-HT+Aβo. (Figure 2E). These results suggest that in SH-SY5Y cells, the protective effect of antidepressants against Aβo-induced cytotoxicity is affected by 5-HT receptor stimulation rather than by α_2_-adrenergic receptor stimulation.

The same experiments were performed with 10 μM WAY-100635 pretreatment, followed by 1 μM antidepressants, in presence of 5 μM Aβo to determine whether the 5-HT receptor mediates the protective effect of antidepressants against Aβo-induced cytotoxicity. Pretreatment with WAY-100635 significantly suppressed the increase in cell viability induced by all three antidepressants (Figure 2F,G). These experiments suggest that the protective effect of antidepressants against Aβo-induced cytotoxicity in SH-SY5Y cells is mediated by 5-HT. 

### 3.2. Effects of Antidepressants on Aβo-Induced Oxidative Stress

Increased oxidative stress has been reported to be an early event in AD because it causes cell membrane damage, mitochondrial dysfunction, and cell death [25,26,27].

#### 3.2.1. Detection of ROS

ROS generation was significantly increased in cells exposed to 5 μM Aβo compared with that in the control or 0.1% DMSO control for 24 h. However, treatment with the Flx + Aβo showed significant suppression of ROS generation caused by Aβo exposure (*p* < 0.0001). Furthermore, treatment with Dlx + Aβo or Mir + Aβo significantly decreased ROS generation compared with Aβo exposure alone (Figure 3A,B). Subsequently, we compared ROS generation with and without the presence of WAY-100635. Pretreatment with WAY-100635 significantly suppressed the reduction in ROS generation induced by treatment with the three antidepressants (Flx, Dlx, or Mir) + Aβo (Figure 3A,B).

The ROS levels of cells exposed to 5 μM Aβo and 5 μM Aβo + 0.1% DMSO were set at 100% to compare the three antidepressants. The ROS levels of Flx + Aβo treatment were 50.17 ± 3.20%; Dlx + Aβo, 39.626 ± 4.14%; and Mir + Aβo, 52.76 ± 3.03%. The degree of suppression in ROS generation by the antidepressant treatment in Aβo-exposed cells was Dlx > Flx = Mir.

The fluorescence microscopy images are shown in Figure 3C–O. Exposure of SH-SY5Y cells to 5 μM Aβo enhanced green fluorescence (Figure 3E,F), but treatment with the three antidepressants + Aβo reduced fluorescence (Figure 3J–L). In addition, pretreatment with WAY-100635 was observed to enhance the fluorescence of the cells treated with the three antidepressants (Flx, Dlx, and Mir) + Aβo (Figure 3M–O).

#### 3.2.2. Detection of Mitochondrial ROS (mito-ROS)

As shown in Figure 4, the levels of mito-ROS in SH-SY5Y cells exposed to Aβo were significantly increased compared to those in the control. Treatment with the three antidepressants + Aβo significantly suppressed mito-ROS generation compared with Aβo exposure alone (n = 10, Tukey’s test).

Mito-ROS levels of cells exposed to 5 μM Aβo and 5 μM Aβo + 0.1% DMSO were set at 100% to compare the effect of the three antidepressants. The mito-ROS levels of Flx + Aβo treatment were 71.96 ± 3.90%; Dlx + Aβo, 54.58 ± 2.10%; and Mir + Aβo, 60.37 ± 3.55%. The extent of suppression in mito-ROS generation by the antidepressant treatment in Aβo-exposed cells was Dlx > Mir > Flx.

We pretreated the cells with WAY-100635 for 1 h. Then, we performed the same experiment with the three antidepressants to determine whether WAY-100635 mediated the inhibitory effects of the three antidepressants on mito-ROS induced by Aβo.

Subsequently, we compared mito-ROS generation with and without the presence of WAY-100635. Pretreatment with WAY-100635 significantly suppressed the inhibitory effect of Dlx + Aβo treatment on mito-ROS generation. 5-HT receptor antagonist pretreatment tended to inhibit the reduction in mito-ROS by Mir + Aβo treatments, although not significantly. However, Flx + Aβo treatment did not change the mito-ROS levels after WAY-100635 pretreatment.

#### 3.2.3. Phospholipid Peroxidation in Cell Membranes

Aβo are believed to permeabilize cellular membranes, causing lipid peroxidation and damaging the structure of the phospholipid bilayer [28]. Phospholipid peroxidation in cell membrane was significantly increased in cells exposed to 5 μM Aβo compared to control cells. Treatment with the three antidepressants significantly reduced Aβo-induced peroxidation of membrane phospholipids.

The cell membrane phospholipid peroxidation levels of cells exposed to 5 μM Aβo and 5 μM Aβo + 0.1% DMSO were set at 100% to compare the three antidepressants. The cell membrane phospholipid peroxidation levels of cells treated with Flx + Aβo were 92.98 ± 0.95%; Dlx + Aβo, 77.16 ± 2.55%; and Mir + Aβo, 78.01 ± 3.80%. The extent of suppression in lipid peroxidation by the antidepressant treatment in Aβo-exposed cells was Dlx = Mir > Flx.

Subsequently, we compared phospholipid peroxidation with and without the presence of WAY-100635. The three antidepressants + Aβo treatments did not change the peroxidation of membrane phospholipids after WAY-100635 pretreatment (Figure 5A,B).

#### 3.2.4. Human HO-1 Levels

The levels of HO-1 in SH-SY5Y cells exposed to 5 μM Aβo were unchanged compared with those in the control. Treatment with Flx + Aβo or Mir + Aβo did not show statistically significant differences. However, compared with 5 μM Aβo, treatment with Dlx + Aβo significantly increased HO-1 levels. Furthermore, Mir treatment tended to increase HO-1 levels compared to controls (Aβo + 0.1% DMSO).

The HO-1 levels of cells exposed to 5 μM Aβo and 5 μM Aβo + 0.1% DMSO were set at 100% to compare the three antidepressants. The HO-1 levels of Flx + Aβo treatment were 116.62 ± 20.77%; Dlx + Aβo, 169.97 ± 23.44%; and Mir + Aβo, 146.38 ± 18.78%. The elevated levels of HO-1 caused by the antidepressant treatment in Aβo-exposed cells were in the following order: Dlx > Mir > Flx. Moreover, pretreatment with WAY-100635 did not affect HO-1 levels induced by treatment with Dlx + Aβo (Figure 6A,B).

#### 3.2.5. Manganese Superoxide Dismutase (MnSOD) Levels

Superoxide dismutase, a major antioxidant enzyme, removes superoxide radicals and protects cells from ROS. MnSOD levels were significantly decreased in cells exposed to Aβo compared with those in the control cells. Treatment with the three antidepressants + Aβo significantly increased MnSOD levels compared with Aβo exposure.

The MnSOD levels of cells exposed to 5 μM Aβo and 5 μM Aβo + 0.1% DMSO were set at 100% to compare the three antidepressants. The MnSOD levels of Flx + Aβo treatment were 145.54 ± 5.66%; Dlx + Aβo, 150.74 ± 7.37%; and Mir + Aβo, 151.29 ± 7.17%. The trend of the elevated levels of MnSOD caused by the antidepressant treatment in Aβo-exposed cells was Dlx = Mir > Flx.

Subsequently, we compared MnSOD levels with and without the presence of WAY-100635. Pretreatment with WAY-100635 suppressed the MnSOD increase of Dlx + Aβo and Mir + Aβo but did not suppress the MnSOD increase for Flx + Aβo treatment (Figure 7A,B).

## 4. Discussion

In this study, we systematically investigated the protective effects of the antidepressants Flx, Dlx, and Mir against the neurotoxicity induced by Aβo, particularly focusing on oxidative stress in SH-SY5Y cells.

In a previous study, when SSRIs (Flx, citalopram) were administered to APP/PS1 (Amyloid Precursor Protein/Presenilin 1) transgenic mice in in vivo experiments, the levels of Aβ in the brain interstitial fluid decreased by 25% [29]. SSRIs increased α-secretase activity (non-amyloidogenic pathway) and reduced β-secretase and γ-secretase cleavage of APP, leading to the production of amyloidogenic Aβ. However, the mechanisms underlying these effects are not fully understood. Moreover, there are few reports on the effects of SNRIs and NaSSAs. In this study, focusing on oxidative stress, we investigated the cellular responses to Flx, Dlx, and Mir regarding ROS production in response to Aβo exposure, membrane lipid peroxidation, mitochondrial ROS production, HO-1 activity, and Mn-SOD.

Aβo induced a decrease in cell viability and an increase in oxidative stress. Aβ is known to undergo aggregation from soluble monomers to form higher-molecular-weight species, including oligomers, protofibrils, and mature fibrils. Among these species, oligomers are recognized as the most toxic forms of Aβ, leading to synaptic loss and neuronal cell death in the brains of individuals with AD. Our previous studies showed that high-molecular-weight Aβ exhibited greater cytotoxicity than low-molecular-weight Aβ [4]. Indeed, neurodegeneration, neuroinflammation, and impaired synaptic plasticity are at least partially attributed to oxidative stress induced by Aβo [30]. Aβo have been demonstrated to induce oxidative stress in neuronal cells through various mechanisms. In the early stages of Aβo formation, oxidative stress from the external environment damages the cellular membrane lipid bilayer structure. Additionally, Aβo induce the generation of ROS primarily within the cells, particularly in the mitochondria [31,32]. Aβo, when administered externally, initially come into contact with the cell membrane. Therefore, it is believed that some of the oxidative stress observed in short-term Aβo exposure experiments may be associated with damage to the cell membrane. Consistent with this hypothesis, short-term exposure to Aβo induces peroxidation of cell membrane phospholipids (Figure 5). Furthermore, Aβo exposure increased the generation of both ROS and mito-ROS (Figure 3 and Figure 4). Oxidative stress occurs when the scavenging capacity of antioxidants is overwhelmed, leading to cellular toxicity [33]. The initiation and progression of many chronic diseases, including diabetes, cancer, and neurodegenerative disorders, are associated with excessive ROS, and oxidative stress is considered one of the causes of cellular damage [34]. In AD, excessive ROS generation and oxidative stress are associated with increased production and/or aggregation of Aβ. This association exacerbates oxidative damage to neuronal cells and contributes to the neuronal cell death observed in AD brains [31].

Flx, when administered for an extended period to rats, has been reported to increase the levels of 2-thiobarbituric acid reactive substances [35]. Even short-term administration to mice has been shown to increase the generation of ROS [36]. Conversely, in non-transgenic animal models of AD, Flx administration hindered the overexpression of inducible nitric oxide synthase (iNOS) and NADPH oxidase 2 (NOX2) induced by Aβo and demonstrated antioxidant effects [37]. This study found that Flx has certain neuroprotective effects against Aβo, which involve antioxidant effects (Table 1). However, the expression of HO-1, induced when cells are exposed to ROS and heavy metals, did not show a significant impact. Flx may weakly affect the regulation of HO-1 induction through the Keap1-Nrf2 pathway (the principal protective response to oxidative and electrophilic stresses) in response to oxidative stress. On the contrary, lower concentrations of Dlx and Mir were required for the inhibition of Aβo-induced cytotoxicity in the MTT assay compared with Flx (Figure 2C,D). Additionally, Dlx and Mir more effectively suppressed the increase in ROS, mito-ROS, and membrane lipid peroxidation levels induced by Aβo compared with Flx (Figure 3, Figure 4 and Figure 5). Furthermore, the antioxidant activities measured by MnSOD and HO-1 levels increased more significantly with Dlx and Mir than with Flx (Figure 6 and Figure 7). The protective effects of Dlx against oxidative stress observed in our study are supported by research blocking rotenone-induced cell death with Dlx pretreatment [38] and reducing cell death induced by hydrogen peroxide in PC12 cells [39]. Furthermore, the protective effect of Mir was demonstrated by increasing Nrf2 in endothelial cells exposed to lipopolysaccharide (LPS) [40]. Additionally, Mir was demonstrated to exert its protective effects by regulating the mRNA levels of several antioxidant enzymes in human monocytes [41]. In the Aβo-induced cell injury observed in this experiment, all three antidepressants, as reported previously, demonstrated antioxidant activity and exhibited cytoprotective effects.

In this study, the treatment concentrations of the three antidepressants were set at 1.0 μM, a concentration not significantly different from the blood concentrations observed in clinical use. In clinical use of Flx, the serum concentration (Cmax) after administration of 20 mg/dose was 16.25 ng/mL (0.047 μM), respectively [42]. When administered as a single dose, the Cmax of Dlx was 24.6 ng/mL (0.083 μM) for the 40 mg dose and 40.1 ng/mL (0.135 μM) for the 60 mg dose [43]. When administered at the typical dose range of 15–45 mg, the blood concentration of Mir is generally around 30–120 ng/mL (0.113–0.452 μM) [44]. In a previous report, exposure of N2 and C17.2 cells to a high concentration of Dlx (100 μM) resulted in significant cytotoxicity [45]. However, studies with 1–5 μM Dlx treatment for 24 h in human cell lines inducing rotenone cytotoxicity [39] and various cell types, including PC12 cells, demonstrated neuroprotective effects at lower concentrations.

These three antidepressants inhibit the serotonin transporter (SERT) and demonstrate antidepressant effects. However, the presence of 5-HT in the synapse significantly influences Aβo-induced neuronal damage (Figure 2E). In AD and mild cognitive impairment (MCI), degeneration of the serotonin system has been observed. In transgenic amyloid mouse models, degeneration of the serotonin system is detected before extensive deposition of cortical Aβ, suggesting the possibility of serotonin system degeneration in preclinical AD. 5-HT is a neurotransmitter and is not typically considered an antioxidant. However, some research findings have reported a relationship between oxidative stress and 5-HT in depression. In vitro experiments have shown that 5-HT exhibits lipid peroxidation inhibition and free radical scavenging effects in the presence of linoleic acid [46,47], and in vivo experiments have demonstrated a significant increase in oxidative stress in 5-HT-deficient mice [48]. In addition, treating rat organotypic raphe slices with various antidepressants for 4 days resulted in a significant increase in extracellular 5-HT concentration. Furthermore, upon comparing the augmentation of 5-HT release, the elevation in extracellular 5-HT levels was approximately twice as high with Dlx compared with that with Flx [49]. In this study, the observed stronger protective effect of Dlx compared with that of Flx against Aβo-induced cellular damage is believed to be attributed to the difference in extracellular 5-HT concentration.

The 5-HT_1A_ receptor is widely expressed in the brain, predominantly being found in regions such as the hippocampus, nasal septum, amygdala, and cortical periphery, serving as both autoreceptors and heteroreceptors. The 5-HT_1A_ receptor has become a therapeutic target for psychiatric disorders, including depression, anxiety disorders, and schizophrenia, because of its presence in key brain areas. Notably, the 5-HT_1A_ heteroreceptor plays a central role in antidepressant effects [50]. Although various antidepressants operate through different mechanisms, they commonly exert their antidepressant effects by increasing the binding of serotonin to the 5-HT_1A_ heteroreceptor. In previous studies on SSRIs, it has been demonstrated that SSRI treatment leads to the binding of 5-HT to the 5-HT_1A_ heteroreceptor, inducing multiple signaling pathways that promote synaptic formation and exhibit neuroprotective effects [51,52]. Furthermore, the high expression of 5-HT_1A_ in the hippocampus indicates its role in the memory process. Particularly, alterations in the expression of hippocampal 5-HT_1A_ receptors during the prodromal stages of AD have sparked discussions regarding whether the regulation of hippocampal 5-HT_1A_ receptors is beneficial or detrimental to the progression of AD [53]. Additionally, a decrease in 5-HT_1A_ receptor binding has been associated with AD, interpreted as a consequence of neuronal loss. It has been observed that [^11^C]WAY-100635 binding is reduced in the hippocampus and medial temporal cortex in mild patients with AD using positron emission tomography imaging [54]. In this study, the neuroprotective effects demonstrated by the three antidepressants against Aβo-induced neuronal damage were suppressed by pretreatment with the 5-HT_1A_ receptor antagonist, WAY-100635. Furthermore, the antidepressant treatment showed antioxidative effects against Aβo-induced oxidative stress, which were attenuated by pretreatment with WAY-100635 (Figure 3). These findings suggest the involvement of antioxidative effects mediated by the 5-HT_1A_ receptor as one of the mechanisms for the neuroprotective effects of antidepressant treatment.

Overall, our investigation demonstrated the protective effects of antidepressants Flx, Dlx, and Mir against Aβo-induced neuronal damage and oxidative stress. This study provides evidence that the neuroprotective effects of these three antidepressants in vitro are induced by antioxidative stress, highlighting the potential involvement of the 5-HT_1A_ receptor as one of the mechanisms for neuroprotection. Antipsychotic medications with 5-HT_1A_ receptor agonist properties or antidepressants with indirect 5-HT_1A_ receptor agonist properties may hold therapeutic promise in targeting the 5-HT_1A_ receptor in the treatment of AD. Given that depression is a significant risk factor for AD, our findings suggest the importance of routine clinical practices such as administering antidepressants early to patients with depression, which may favorably impact the prevention of AD onset. However, considering the direct antioxidative effects demonstrated by the antidepressants that do not act through the 5-HT_1A_ receptors, further detailed research on the antioxidative actions of antidepressants is warranted. Additionally, more extensive studies are needed to explore the effects of Aβo on 5-HT receptors.

Limitations of this study need to be discussed. Although we employed Aβo-induced neuronal injury as the representative pathology of neuronal injury in AD disease, differences between in vitro models and human disease need to be highlighted. Our study examined only the neurons and did not include the effects of the antidepressant drugs on microglia or astrocytes, which need to be examined.

## 5. Conclusions

The antidepressants Flx, Dlx, and Mir demonstrated neuroprotective effects through multiple mechanisms. It is evident that the mechanism of action involves the 5-HT_1A_ receptor, and the promotion of antioxidant effects may be part of the neuroprotective mechanism of the 5-HT_1A_ receptor. The findings of this study suggest that early use of antidepressants in patients with depression may act preventatively against potential AD onset, emphasizing the contribution of antidepressants to AD prevention. While numerous unresolved issues remain to be tackled, administering antidepressants to pre-symptomatic or early-stage AD patients is suggested to be beneficial.

## Figures and Tables

**Figure 1 biomedicines-12-01158-f001:**
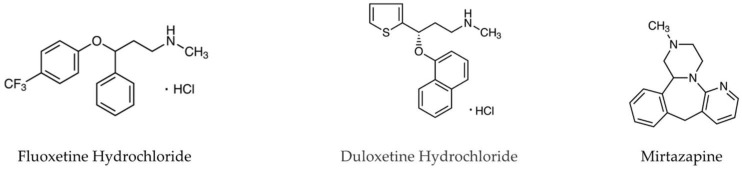
Structures of fluoxetine, duloxetine, and mirtazapine.

**Figure 2 biomedicines-12-01158-f002:**
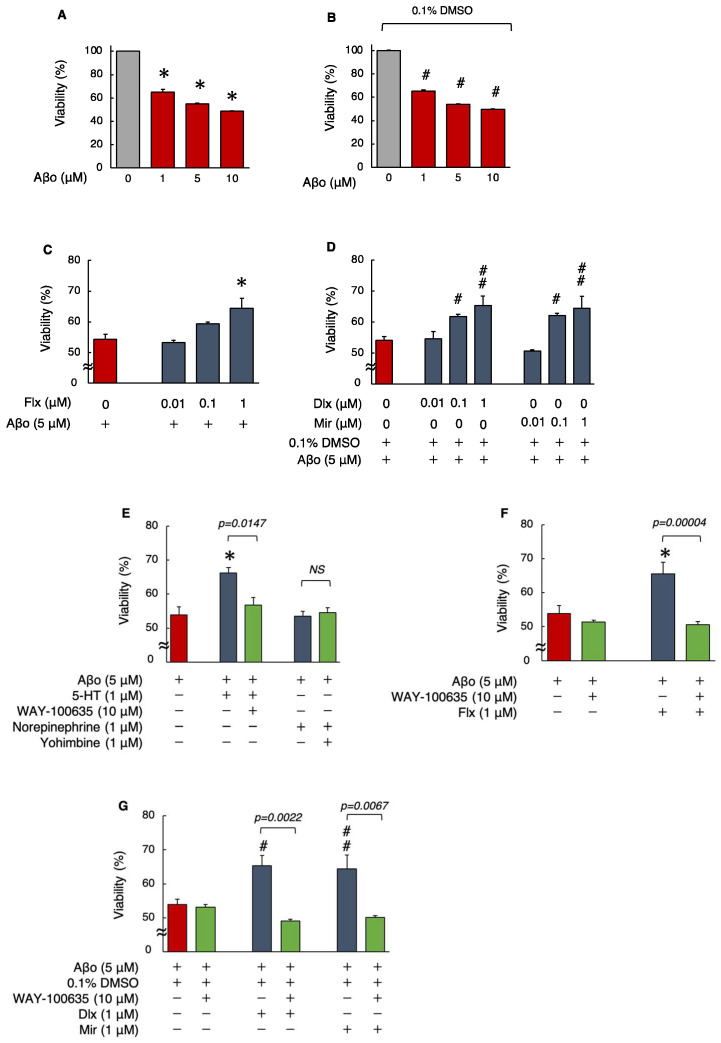
Effect of the antidepressants on cell viability in Aβo-stimulated SH-SY5Y cells. (**A**,**B**) Cell viability of SH-SY5Y cells exposed to Aβo (1, 5, and 10 μM) for 24 h. Each value is expressed relative to the viability of control (**A**) or 0.1% DMSO-treated (**B**) cells (set to 100%). Each value is expressed as mean ± SEM. The *p*-value in ANOVA was <0.0001. *, *p* < 0.0001, versus control cells. #, *p* < 0.0001 versus control cells with 0.1% DMSO (*n* = 6, Dunnett’s test). (**C**,**D**) Cell viability of SH-SY5Y cells exposed to 5 μM Aβo and treated simultaneously with the antidepressants (0.01–1 μM) for 24 h. Each value is expressed as mean ± SEM. The *p*-values in ANOVA were < 0.001. *, *p* < 0.01 versus 5 μM Aβo-exposed cells. #, *p* < 0.05; ##, *p* < 0.005 versus 5 μM Aβo + 0.1 % DMSO-exposed cells (*n* = 10, Dunnett’s test). In the absence of 5 μM Aβo, cells treated with 1 μM Flx, Dlx, and Mir were 99.7 ± 0.93, 99.9 ± 0.11, and 98.58 ± 0.86% viable, respectively (no significant difference, *n* = 10, Dunnett’s test). (**E**) SH-SY5Y cells were pretreated with a 5-HT antagonist (WAY-100635) and α_2_-adrenergic receptor antagonist (yohimbine) and with Aβo + 5-HT or Aβo + NE. The *p*-values in ANOVA were <0.0001. Results are expressed as means ± SEMs. *, *p* < 0.0005, versus 5 μM Aβo-exposed cells (*n* = 10, Tukey’s test). In the absence of 5 μM Aβo, cells treated with 10 μM WAY-100635 and 1 μM yohimbine were 99.6 ± 1.05% and 98.9 ± 1.37% viable, respectively (no significant difference, *n* = 10, Tukey’s test). (**F**,**G**) SH-SY5Y cells were treated with the three antidepressants + Aβo for 24 h after WAY-100635 pretreatment for 1 h. The *p*-values in ANOVA were <0.0001. Results are expressed as means ± SEMs. *, *p* < 0.01 versus 5 μM Aβo-exposed cells. #, *p* < 0.05; ##, *p* < 0.01 versus 5 μM Aβo + 0.1% DMSO-exposed cells (*n* = 10, Tukey’s test). Aβo, Aβ oligomer; DMSO, dimethyl sulfoxide; Flx, fluoxetine; Dlx, duloxetine; Mir, mirtazapine; 5-HT, 5-hydroxytryptamine; NS, not significant; SEM, standard error of the mean; ANOVA, analysis of variance.

**Figure 3 biomedicines-12-01158-f003:**
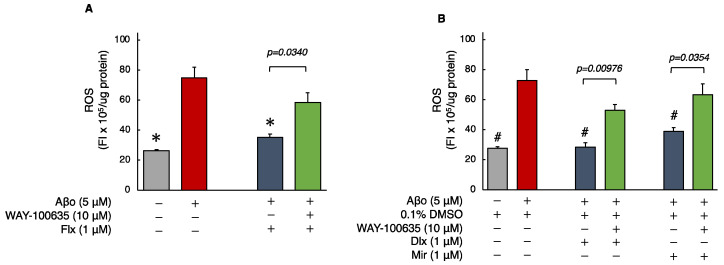
Effect of the antidepressants on ROS generation in Aβo-stimulated SH-SY5Y cells. (**A**,**B**) SH-SY5Y cells were treated with the three antidepressants + Aβo for 24 h after WAY-100635 pretreatment for 1 h. The *p*-value in ANOVA was <0.0001. Results are expressed as means ± SEMs. *, *p* < 0.0001, versus Aβo-exposed cells. #, *p* < 0.0001, versus Aβo + 0.1% DMSO-exposed cells. In the absence of 5 μM Aβo, ROS levels of Flx-, Dlx-, and Mir-treated cells were 29.6 ± 1.44 (*p* = 0.9927 vs. control), 27.7 ± 1.25 (*p* = 1.0000 vs. 0.1% DMSO control) and 23.9 ± 1.89 (*p* = 0.9997 vs. 0.1% DMSO control) fluorescence intensity × 10^6^/μg protein (no significant difference, *n* = 10, Tukey’s test). (**C**–**O**) SH-SY5Y cells were observed using fluorescence microscopy. The scale bars represent 100 μm. Aβo, Aβ oligomer; DMSO, dimethyl sulfoxide; ROS, reactive oxygen species; Flx, fluoxetine; Dlx, duloxetine; Mir, mirtazapine; SEM, standard error of the mean; ANOVA, analysis of variance.

**Figure 4 biomedicines-12-01158-f004:**
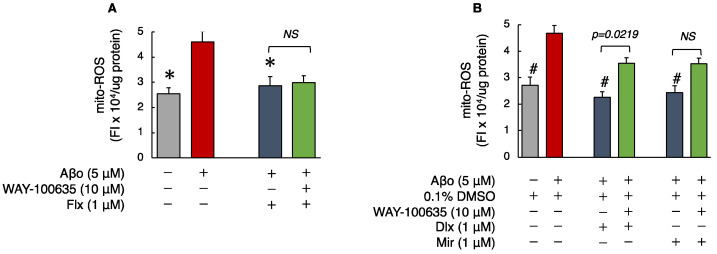
Effect of the antidepressants on mito-ROS generation in Aβo-stimulated SH-SY5Y cells. (**A**,**B**) SH-SY5Y cells were treated with the three antidepressants + Aβo for 24 h after WAY-100635 pretreatment for 1 h. The *p*-value in ANOVA was 0.005. Each value expresses the mean ± SEM of 10 individually treated samples (Tukey’s test). *, *p* < 0.005, versus Aβo-exposed cells. #, *p* < 0.0001, versus Aβo + 0.1% DMSO-exposed cells. In the absence of 5 μM Aβo, mito-ROS levels of Flx-, Dlx-, and Mir-treated cells were 2.16 ± 0.15, 2.15 ± 0.16, 2.55 ± 0.11, and 2.39 ± 0.13 fluorescence intensity × 10^4^/μg protein (no significant difference, *n* = 10, Tukey’s test). Aβo, Aβ oligomer; DMSO, dimethyl sulfoxide; mito-ROS, mitochondrial reactive oxygen species; Flx, fluoxetine; Dlx, duloxetine; Mir, mirtazapine; NS, not significant; SEM, standard error of the mean; ANOVA, analysis of variance.

**Figure 5 biomedicines-12-01158-f005:**
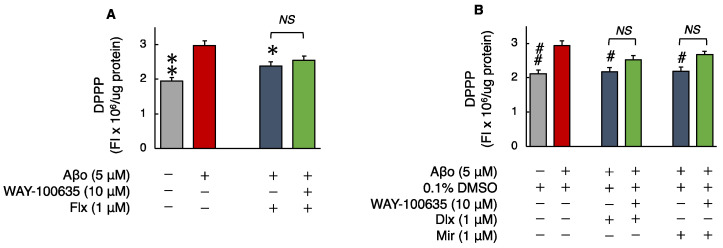
Effect of the antidepressants on membrane phospholipid peroxidation in Aβo-stimulated SH-SY5Y cells. (**A**,**B**) SH-SY5Y cells were treated with the three antidepressants + Aβo for 30 min after WAY-100635 pretreatment for 10 min. The *p*-value in ANOVA was 0.0009. Each value expresses the mean ± SEM of 10 individually treated samples (Tukey’s test). *, *p* < 0.005, **, *p* < 0.001 versus Aβo and #, *p* < 0.05; ##, *p* < 0.005, versus Aβo + 0.1% DMSO-exposed cells. In the absence of 5 μM Aβo, membrane phospholipid peroxidation levels of Flx-, Dlx-, and Mir-treated cells were 1.85 ± 0.06, 1.90 ± 0.06, and 1.86 ± 0.07 fluorescence intensity × 10^4^/μg protein (no significant difference, *n* = 10, Tukey’s test). Aβo, Aβ oligomer; DMSO, dimethyl sulfoxide; DPPP, diphenyl-1-pyrenylphosphine; Flx, fluoxetine; Dlx, duloxetine; Mir, mirtazapine; NS, not significant; SEM, standard error of the mean; ANOVA, analysis of variance.

**Figure 6 biomedicines-12-01158-f006:**
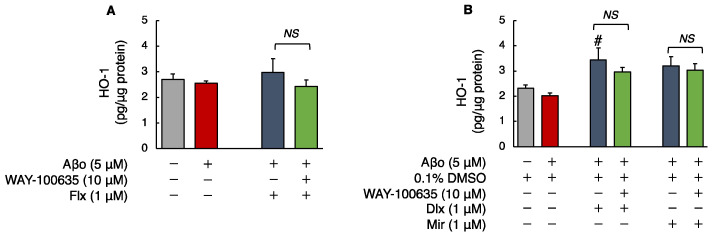
Effect of the antidepressants on HO-1 levels in Aβo-stimulated SH-SY5Y cells. (**A**,**B**) SH-SY5Y cells were treated with the three antidepressants + Aβo for 24 h after WAY-100635 pretreatment for 1 h. The *p*-value in ANOVA was 0.0150. Each value expresses the mean ± SEM of 6 individually treated samples (Tukey’s test). #, *p* < 0.05, versus Aβo + 0.1% DMSO-exposed cells. In the absence of 5 μM Aβo, HO-1 levels of Flx-, Dlx-, and Mir-treated cells were 3.27 ± 0.29, 3.61 ± 0.34, and 3.20 ± 0.48 fluorescence intensity × 10^4^/μg protein (no significant difference, *n* = 6, Tukey’s test). Aβo, Aβ oligomer; DMSO, dimethyl sulfoxide; HO-1, heme oxygenase 1; Flx, fluoxetine; Dlx, duloxetine; Mir, mirtazapine; NS, not significant; SEM, standard error of the mean; ANOVA, analysis of variance.

**Figure 7 biomedicines-12-01158-f007:**
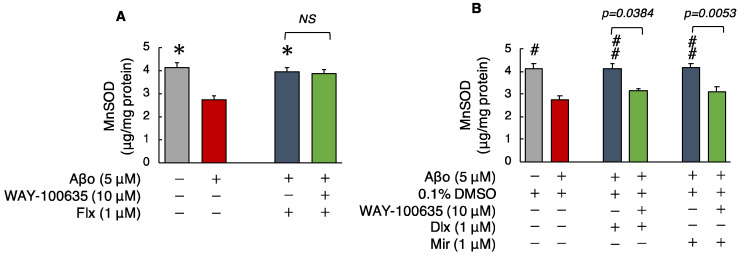
Effect of the antidepressants on MnSOD levels in Aβo-stimulated SH-SY5Y cells. (**A**,**B**) SH-SY5Y cells were treated with the three antidepressants + Aβo for 24 h after WAY-100635 pretreatment for 1 h. The *p*-value in ANOVA was <0.0001. Each value expresses the mean ± SEM of 10 individually treated samples (Tukey’s test). *, *p* < 0.001, versus Aβo-exposed cells. #, *p* < 0.005; ##, *p* < 0.001, versus Aβo + 0.1% DMSO-exposed cells. In the absence of 5 μM Aβo, Mn-SOD levels of Flx-, Dlx-, and Mir-treated cells were 4.59 ± 0.79, 4.07 ± 0.36, and 4.48 ± 0.32 fluorescence intensity × 10^4^/μg protein (no significant difference, *n* = 10, Tukey’s test). Aβo, Aβ oligomer; DMSO, dimethyl sulfoxide; MnSOD, manganese superoxide dismutase; Flx, fluoxetine; Dlx, duloxetine; Mir, mirtazapine; NS, not significant; SEM, standard error of the mean; ANOVA, analysis of variance.

**Table 1 biomedicines-12-01158-t001:** Pharmacological activities.

Activities	Measurements	Flx	Dlx	Mir
Viability	MTT	+	++	++
Antioxidant effects	ROS	+	++	+
	Mito-ROS	+	++	++
	Phospholipid peroxidation	+	+	+
Activation of antioxidative enzyme activity	HO-1	−	+	−
	Mn-SOD	+	+	+

++: strong intensity reaction, +: weak intensity reaction, −: no change. MTT, 3-(4, 5-dimethylthiazoyl-2-yl)-2-5, diphenyltetrazolium bromide; ROS, reactive oxygen species; mito-ROS, mitochondrial-ROS; HO-1, heme oxygenase 1; MnSOD, manganese superoxide dismutase; Flx, fluoxetine; Dlx, duloxetine; Mir, mirtazapine.

## Data Availability

The original contributions presented in the study are included in the article/Appendix A, further inquiries can be directed to the corresponding author/s.

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
