# Peer review of "Comparison of Protective Effects of Antidepressants Mediated by Serotonin Receptor in Aβ-Oligomer-Induced Neurotoxicity"

_biomedicines, 2024, doi:10.3390/biomedicines12061158_

Round 1
Reviewer 1 Report
Comments and Suggestions for Authors
Ref: biomedicines-2956198
This is an elegant experimental study, investigating the possible role of antidepressants in Alzheimer’s disease. The authors used Aβ oligomers to induce toxic damage by oxidative stress and found that antidepressants increased cell viability and decreased oxidative stress induced by Aβ oligomers. Pretreatment with 5-HT1A antagonist suppressed the neuroprotective effects of antidepressants. Overall, the results support the hypothesis that treatment with antidepressants may have a neuroprotective role in Alzheimer’s disease and this role is related to antioxidant action and somehow mediated by 5-HT1A receptors.
The study is well designed and performed. Experimental and statistical methods are appropriate and reference are up to date.
Depression may occur as a symptom of Alzheimer’s disease, but it may also be a risk factor of this disorder. Thus, treatment of depression may protect from hippocampal atrophy and Alzheimer’s disease. The present study provides new data on the possible interplay between depression and Aβ-induced neurodegeneration and the possible neuroprotective effects of antidepressants.
I would only suggest to expand the discussion on depression as a risk factor of Alzheimer’s disease and the significance of these findings in every-day clinical practice (early treatment of depression etc)
Author Response
Thank you very much for your valuable feedback
>I would only suggest to expand the discussion on depression as a risk factor of Alzheimer’s disease and the significance of these findings in every-day clinical practice (early treatment of depression etc)
Taking your points into consideration, we have added additional discussion to address them. We appreciate your continued support.
Reviewer 2 Report
Comments and Suggestions for Authors
1. In general, the authors divided paragraphs in many individual sentences. These writing does not look organized. Please re-organize the entire writing. Moreover, why the authors presented experimental results in two figures; one for Flx and the other for Dlx and Mir. Can the figures be combined?
2. For the preparation of Abeta oligomers,
a. the authors incubated monomeric Abeta for 1 h at 37 degree, however, the characterization of those oligomers should be performed.
b. Also, can the authors sure that all the oligomeric species are in same conditions for the experiment?
c. From what size, can we consider the “HMW oligomer”?
3. For the MTT assay,
a. why did the authors add 0.1% DMSO to the cells?
b. In figure 2a, without Abeta oligomers (control sample), why the cell viability is not 100%? How the authors determine the viability of compounds treated cells?
c. Why the 5 uM Abeta oligomer-treated cell viability are different in Figure 2a and 2c?
d. Why the authors treated 5-HT, NE, and WAY-100635 to the cells? It is not clearly mentioned in the manuscript.
e. Are the compounds stable for 24 h? If not, it should be examine what species are existed in the solution.
4. In Figure 3,
a. The fluorescence signal from Figure 3G, 3H, and 3I indicates that Flx, Dlx, and Mir could enhance the ROS generation. In table 1, however, the authors mentioned that all three compounds could reduce the ROS production. It should be discussed in more detail.
b. Based on Figure 3H and 3K, it is difficult to conclude that Dlx could reduce the production of ROS which is different conclusion from the authors in line 345-346.
5. In Figure 7,
a. The levels of MnSOD with and without Flx, Dlx, and Mir look similar. However, the authors mentioned “… antidepressants treatments in Abo-exposed cells was Dlx = Mir > Flx …” in line 454-455. This sentence is wrong based on the results.
b. Moreover, the authors described the results in Figure 7 are the “activity” of MnSOD. However, in the text, the authors mentioned the results as “levels”. It does not match.
Comments on the Quality of English LanguageIn general, the authors divided paragraphs in many individual sentences. These writing does not look organized. Please re-organize the entire writing.
Author Response
Thank you very much for your pertinent and productive comments. We agree to your following comments and amended our manuscript accordingly. We are very sorry we had several typing and English usage errors, and are very thankful to for your careful reading.
- In general, the authors divided paragraphs in many individual sentences. These writing does not look organized. Please re-organize the entire writing. Moreover, why the authors presented experimental results in two figures; one for Flx and the other for Dlx and Mir. Can the figures be combined?
Thank you very much for your feedback. We have thoroughly revised the entire text. Additionally, we divided Flx and Dlx/Mir into two separate figures because Flx dissolves in the medium, while Dlx and Mir do not, they were dissolved in DMSO. The final concentration of DMSO was adjusted to 0.1%. Consequently, for Flx, cells cultured in medium alone were used as controls, whereas for Dlx and Mir, cells cultured in medium containing 0.1% DMSO were used as controls. Due to the differing control comparisons, we opted to present the data in separate graphs for clarity.
- For the preparation of Abeta oligomers,
- the authors incubated monomeric Abeta for 1 h at 37 degree, however, the characterization of those oligomers should be performed.
- Also, can the authors sure that all the oligomeric species are in same conditions for the experiment?
- From what size, can we consider the “HMW oligomer”?
Thank you very much for your thoughtful comment. As indicated in the reference, we have been using the same adjustment method as before. The adjusted Aβ oligomers are confirmed by both peak identification through SEC and visual confirmation via electron microscopy, as depicted in Supplemental Figure 1. These oligomers are understood as assemblies of tens to hundreds of monomers.
- For the MTT assay,
- why did the authors add 0.1% DMSO to the cells?
As mentioned in the response to Comment 1, since Dlx and Mir do not dissolve in medium, they were dissolved in DMSO. Therefore, to maintain a consistent DMSO concentration of 0.1% when treating cells with Dlx and Mir, cells were also treated with 0.1% DMSO as a control group alongside Dlx and Mir-treated cells.
- In figure 2a, without Abeta oligomers (control sample), why the cell viability is not 100%? How the authors determine the viability of compounds treated cells?
First, we set the absorbance value of cells cultured in 10% FBS-containing medium to represent 100% viability, and the absorbance value of cells treated with 0.1% saponin to represent 0% viability. Since cells cultured in serum-free medium showed a slight decrease in viability compared to those cultured in 10% FBS-containing medium, the viability of the control in Figure 2a is no longer 100%.
- Why the 5 uM Abeta oligomer-treated cell viability are different in Figure 2a and 2c?
The viability of cells treated with 5μM Aβ in Fig 2a and Fig 2c were 54.7% and 50.9%, respectively. We attribute the slight discrepancy to the fact that the experiments for Fig 2a and Fig 2c were conducted on different days.
- Why the authors treated 5-HT, NE, and WAY-100635 to the cells? It is not clearly mentioned in the manuscript.
We investigated the direct effects of administering 5-HT and NE by directly treating them, as synaptic levels of 5-HT increase with Flx treatment and both 5-HT and NE increase with Dlx and Mir treatments. These findings have been duly noted in the main text.
- Are the compounds stable for 24 h? If not, it should be examine what species are existed in the solution.
Based on referencing these prior studies, it is considered that each compound remains stable for at least 24 hours.
(1)Am J Hosp Pharm. 1994 May 15;51(10):1342-5. Stability of fluoxetine hydrochloride in fluoxetine solution diluted with common pharmaceutical diluents.
Fluoxetine hydrochloride was found to remain stable for 8 weeks when diluted to 1 or 2 mg/mL in a common pharmaceutical diluent and stored at either 5℃ or 30℃.
(2) Journal of Taibah University for Science. Volume 8, Issue 4, October 2014, Pages 357-363. Development and validation of an analytical method for the stability of duloxetine hydrochloride.
Duloxetine remained stable under neutral hydrolysis (water, 4 hours). Additionally, it was stable against decomposition at room temperature for 24 hours with 10%, 15%, and 30% hydrogen peroxide. However, it appeared to be highly unstable under acidic conditions.
(3) Iranian Journal of Pharmaceutical Research (2014), 13 (3): 853-862. A Rapid and Sensitive HPLC-Fluorescence Method for Determination of Mirtazapine and Its two Major Metabolites in Human Plasma.
The stability of mirtazapine in stock solutions and QC plasma samples (at concentrations of 10, 100, and 500 ng/mL) was evaluated. Under three different conditions (room temperature for 4 weeks, -20℃ for 6 months, and after five freeze-thaw cycles), no significant decrease in the concentrations of mirtazapine and its metabolites was observed. The overall relative standard deviation was less than 5%.
- In Figure 3,
- The fluorescence signal from Figure 3G, 3H, and 3I indicates that Flx, Dlx, and Mir could enhance the ROS generation. In table 1, however, the authors mentioned that all three compounds could reduce the ROS production. It should be discussed in more detail.
- Based on Figure 3H and 3K, it is difficult to conclude that Dlx could reduce the production of ROS which is different conclusion from the authors in line 345-346.
I sincerely apologize for any confusion caused by several similar experiments. We rechecked the images in Figure 3, it has come to our attention that the images for Figure 3 G, H, and I were mistakenly replaced with images from another set (Aβo-exposed cells). We deeply regret this oversight and have promptly corrected the photographs.
- In Figure 7,
- The levels of MnSOD with and without Flx, Dlx, and Mir look similar. However, the authors mentioned “… antidepressants treatments in Abo-exposed cells was Dlx = Mir > Flx …” in line 454-455. This sentence is wrong based on the results.
Thank you very much for your comment. As noted in the results section L464-468, we expressed that the MnSOD levels were 145.54 ± 5.66% for Flx + Aβo treatment, 150.74 ± 7.37% for Dlx + Aβo treatment, and 151.29 ± 7.17% for Mir + Aβo treatment. Consequently, we concluded that Dlx = Mir > Flx.
- Moreover, the authors described the results in Figure 7 are the “activity” of MnSOD. However, in the text, the authors mentioned the results as “levels”. It does not match.
Thank you very much for your feedback. We agree with your comment and corrected the expression in the Figure to "levels".
Reviewer 3 Report
Comments and Suggestions for Authors
Abstract: informative and readable
Introduction:
· I can not agree with the statement: ‘Although the exact mechanism underlying AD remains unknown, prevailing theories point to Aβ as a key factor.’ The amyloid cascade is one of AD factors but there is numerous studies revealeing that the Ab is not crucial for the disease development and cannot be considered as a key factor. Please change the statement.
· Please provide references to the sentences: ‘Tricyclic antidepressants are currently not extensively recommended for patients with major depressive disorder due to serious side effects, with selective serotonin reuptake inhibitors (SSRIs) being the most commonly prescribed.’ and ‘SSRIs are considered safe for long-term use as their side effects are generally well tolerated. However, there is a need to expand research as other classes of antidepressants may need to be prescribed when SSRIs fail.’ and lines 104-109
Methods: well planned, adequate for aim of the studies
Statistical analysis: provided
Results: presented in form of figures along with detail analsis
Discussion: well prepared based on up-to-date references along with dicussion based on obtained results.
Conclusions: can be improved, more details should be provided
References: adequate
Author Response
Thank you very much for taking the time to thoroughly read through our manuscript and providing thoughtful comments.
>I can not agree with the statement: ‘Although the exact mechanism underlying AD remains unknown, prevailing theories point to Aβ as a key factor.’ The amyloid cascade is one of AD factors but there is numerous studies revealeing that the Ab is not crucial for the disease development and cannot be considered as a key factor. Please change the statement.
Thank you for your feedback. As you pointed out, we realized that we had been overly focused on the amyloid cascade hypothesis. We have since revised our wording accordingly (Line 52-54).
>Please provide references to the sentences: ‘Tricyclic antidepressants are currently not extensively recommended for patients with major depressive disorder due to serious side effects, with selective serotonin reuptake inhibitors (SSRIs) being the most commonly prescribed.’ and ‘SSRIs are considered safe for long-term use as their side effects are generally well tolerated. However, there is a need to expand research as other classes of antidepressants may need to be prescribed when SSRIs fail.’ and lines 104-109
Thank you for your feedback. According to the NIH website*, the most commonly prescribed antidepressants are SSRIs. Furthermore, a study comparing SSRIs to tricyclic antidepressants** suggest that SSRIs have higher tolerability. We have added this to the manuscript. (Line 92-94)
* https://magazine.medlineplus.gov/article/commonly-prescribed-antidepressants-and-how-they-work#:~:text=Selective%20serotonin%20reuptake%20inhibitors%20(SSRIs,sertraline%2C%20paroxetine%2C%20and%20escitalopram.
** Anderson IM. Selective serotonin reuptake inhibitors versus tricyclic antidepressants: a meta-analysis of efficacy and tolerability. J Affect Disord. 2000 Apr;58(1):19-36. doi: 10.1016/s0165-0327(99)00092-0. PMID: 10760555.
>Conclusions: can be improved, more details should be provided
Thank you for your feedback. As you pointed out, we agree that the conclusion was somewhat concise. Accordingly, we have made additional revisions to it (Line 605-612).
Round 2
Reviewer 2 Report
Comments and Suggestions for Authors
Although the authors revised the manuscript, still is it not suitable for publication.
1. For the MTT assay,
a. Still I cannot understand. In figure 2a, without Abeta oligomers (control sample), why the cell viability is not 100%? How the authors determine the viability of compounds treated cells?
b. Also, if the 5 uM Abeta oligomer-treated cell viability are different in different days, the results are considered as not reproducible. Then, it cannot make a conclusion.
2. In Figure 3,
a. The compounds, Flx, Dlx, and Mir may enhance the ROS generation compared to the control. It needs more detailed discussion.
b. For me, Figure 3K and 3L look same. It is ethically wrong.
Author Response
- For the MTT assay,
- Still I cannot understand. In figure 2a, without Abeta oligomers (control sample), why the cell viability is not 100%? How the authors determine the viability of compounds treated cells?
- Also, if the 5 uM Abeta oligomer-treated cell viability are different in different days, the results are considered as not reproducible. Then, it cannot make a conclusion.
Thank you very much for this comment. We agree with your comment and recalculated all values and redrawn the graph with control as 100%.
- In Figure 3,
- The compounds, Flx, Dlx, and Mir may enhance the ROS generation compared to the control. It needs more detailed discussion.
Thank you very much for this comment.
For further confirmation, we have reexamined the ROS experiments. Additionally, we have incorporated ROS values into the legends and displayed the results of treatment with the antidepressants alone. Across all three antidepressant treatments, no significant differences were observed between the control or DMSO control groups.
- For me, Figure 3K and 3L look same. It is ethically wrong.
We deeply apologize for our mistake. We truly appreciate your attention to it. We have corrected the images to the accurate ones.
Round 3
Reviewer 2 Report
Comments and Suggestions for Authors
The results shown in Figure 3 is the key finding of the study, however, the authors did make mistake twice.
It reduced the trust of the experimental data presented in this manuscript.
In order to cover up the trust, please provide more than 20 images of each condition in Figure 3.
Author Response
I am writing to express my sincere apologies for the repeated inconvenience caused by our oversight. We deeply regret the trouble this has caused you and are grateful for your feedback on our shortcomings. I have attached more than 20 images for each condition, including those used in my previous submissions. We will take diligent steps to ensure that such issues do not occur in the future.
Thank you for your understanding and cooperation.
Round 4
Reviewer 2 Report
Comments and Suggestions for Authors
My concerns are cleared. The manuscript is now suitable for publication.